# Implementing the CRISPR/Cas9 Technology in *Eucalyptus* Hairy Roots Using Wood-Related Genes

**DOI:** 10.3390/ijms21103408

**Published:** 2020-05-12

**Authors:** Ying Dai, Guojian Hu, Annabelle Dupas, Luciano Medina, Nils Blandels, Hélène San Clemente, Nathalie Ladouce, Myriam Badawi, Guillermina Hernandez-Raquet, Fabien Mounet, Jacqueline Grima-Pettenati, Hua Cassan-Wang

**Affiliations:** 1Laboratoire de Recherche en Sciences Végétales, Université de Toulouse III, CNRS, UPS, UMR 5546, 24 Chemin de Borde Rouge, 31320 Castanet-Tolosan, France; ying.dai@lrsv.ups-tlse.fr (Y.D.); annabelle.dupas@lrsv.ups-tlse.fr (A.D.); luciano.medina@lrsv.ups-tlse.fr (L.M.); nils.blandel@lrsv.ups-tlse.fr (N.B.); san-clemente@lrsv.ups-tlse.fr (H.S.C.); ladouce@lrsv.ups-tlse.fr (N.L.); mounet@lrsv.ups-tlse.fr (F.M.); grima@lrsv.ups-tlse.fr (J.G.-P.); 2UMR 990, Génomique et Biotechnologie des Fruits, Université de Toulouse, INP-ENSA Toulouse, Avenue de l’Agrobiopole, 31326 Castanet-Tolosan, France; guojian.hu@etu.ensat.fr; 3TBI, Université de Toulouse, CNRS, INRAE, INSA, 31400 Toulouse, France; Myriam.Badawi@univ-lemans.fr (M.B.); hernandg@insa-toulouse.fr (G.H.-R.); 4Laboratoire Mer Molécules Santé, MMS EA2160 Le Mans Université, 72085 Le Mans, France

**Keywords:** CRISPR/Cas9, genome editing, cinnamoyl-CoA reductase, Aux/IAA, wood, secondary cell walls, *Eucalyptus*, lignin, hairy roots, FT-IR spectroscopy

## Abstract

Eucalypts are the most planted hardwoods worldwide. The availability of the *Eucalyptus grandis* genome highlighted many genes awaiting functional characterization, lagging behind because of the lack of efficient genetic transformation protocols. In order to efficiently generate knock-out mutants to study the function of eucalypts genes, we implemented the powerful CRISPR/Cas9 gene editing technology with the hairy roots transformation system. As proofs-of-concept, we targeted two wood-related genes: *Cinnamoyl-CoA Reductase1* (*CCR1*), a key lignin biosynthetic gene and *IAA9A* an auxin dependent transcription factor of *Aux/IAA* family. Almost all transgenic hairy roots were edited but the allele-editing rates and spectra varied greatly depending on the gene targeted. Most edition events generated truncated proteins, the prevalent edition types were small deletions but large deletions were also quite frequent. By using a combination of FT-IR spectroscopy and multivariate analysis (partial least square analysis (PLS-DA)), we showed that the *CCR1*-edited lines, which were clearly separated from the controls. The most discriminant wave-numbers were attributed to lignin. Histochemical analyses further confirmed the decreased lignification and the presence of collapsed vessels in *CCR1*-edited lines, which are characteristics of *CCR1* deficiency. Although the efficiency of editing could be improved, the method described here is already a powerful tool to functionally characterize eucalypts genes for both basic research and industry purposes.

## 1. Introduction

Eucalypts are among the leading sources of woody biomass worldwide [1,2]. Due to their rapid growth rate, broad adaptability to diverse edaphoclimatic conditions and their multipurpose wood properties, they are the most planted trees worldwide. Eucalypts wood is currently used in the emerging areas of biofuels and biomaterials in addition to more traditional uses such as pulp and paper production, thereby extending the already considerable economic importance of these trees. Wood is mainly composed of secondary cell walls (SCWs), which contain three major polymers: cellulose, hemicelluloses and lignins. The proportions of each of these polymers and the interactions between them underlie the composition and the structure of the SCWs, which are major determinants of industrial processing efficiencies [3].

The availability of the *Eucalyptus grandis* genome [4] has allowed genome-wide characterization of many gene families, notably those involved in the lignin biosynthetic pathway [5] as well as transcription factor families containing members known to regulate SCW formation such as the R2R3-MYB [6], NAC [7], ARF [8] and Aux/IAA [9] among others. These studies have underscored many new candidates potentially regulating wood formation that need to be functionally characterized. The bottleneck to functionally characterize genes in *Eucalyptus* has always been stable transformation. Although stable transformation protocols have been established for several *Eucalyptus* species [10,11,12,13], they are very time-consuming and present low efficiencies explaining why only very few functional studies have been performed in transgenic eucalypts (reviewed in [11]). To overcome these limitations, we have set up an alternative stable transformation system for *E. grandis* using *Agrobacterium rhizogenes* that allows the development of composite plants with wild-type shoots and transgenic roots easily detectable by fluorescent markers [14]. We have further shown that this system is suitable to elucidate the function of genes involved in xylem or SCW formation particularly important for woody species. As a proof-of-concept, we used the down-regulation of *Cinnamoyl CoA reductase1* (*EgrCCR1*), the penultimate step of the lignin branch pathway through antisense strategy [14].

During the last three decades, antisense RNA, virus-induced gene silencing and RNA interference were the most used methods for gene silencing in plants and provided very useful insights in the function of many genes especially in plants for which no mutant collection was available. However, silencing was not as effective as in mutants since gene function was interrupted indirectly by repressing the corresponding mRNA, often leading to partial repressive effect and in some cases to unpredicted effects. The Clustered Regularly Interspaced Short Palindromic Repeats/CRISPR-associated protein 9 (CRISPR/Cas9) based genome editing that can induce efficiently targeted mutations, revolutionized reverse genetics in all systems and was considered as the breakthrough of year 2015 [15,16]. The CRISPR/Cas9 method is based on the ability of Cas9, a RNA-guided endonuclease from *Streptococcus pyogenes*, to cut a DNA double strand at a specific region [17]. A complementary single-guide RNA (sgRNA) forming a complex with Cas9 will specifically recognize a target DNA region by base-pairing. The Cas9 will then cut directly upstream of a DNA motif of a 2–6 base pair long, called Protospacer Adjacent Motif (PAM). The sequence of the PAM depends from the origin of the Cas9 endonuclease [18].

Currently, CRISPR/Cas9 is the system of choice to targeted mutagenesis in a growing number of plants including woody plants [19]. The possibility to obtain null mutations in the T0 generation is especially important for trees that have very long reproductive cycles [20,21] and like poplar or eucalypts are propagated vegetatively. Another characteristic of outcrossing trees is their high degree of genome heterozygosity and the presence of sequence polymorphisms at the target sites can render CRISPR editing ineffective [19]. The greatest progresses have been made with poplar, the first tree to be genome-edited by CRISPR with high efficiency [20,22] and for which allele-sensitive bioinformatic resources facilitating genome editing in heterozygous species have been developed [21,23]. Due to the importance of wood properties for industrial applications, most of CRISPR gene editing studies in poplar have targeted SCW composition and phenylpropanoid metabolism including lignin [24,25,26,27,28]. A large study encompassing more than 500 transgenic events has also reported successful mutations of essential flowering genes to prevent bisexual fertility [29]. The goal was to produce infertile trees thereby solving potential seed or pollen dispersal, which are of concern for trees that are vegetatively propagated for industrial plantations.

Eucalypts are diploid species (2*n* = 22), each homologous chromosome contains one allele, so each gene has two alleles. Theoretically for each sgRNA, the CRISPR/Cas9 system can introduce up to two types of mutations in the targeted gene except the chimer: (i) if the two alleles are mutated, it is a biallelic mutation, which can be either homozygote mutation (if the two alleles share exactly the same mutation), or biallelic (if the mutations are different in the two alleles) and (ii) if only one of the alleles is mutated, it is a monoallelic mutation, which does not theoretically produce full knock-out mutations because a wild-type allele is remaining, but it may produce knock-down mutations. However, in the case of chimera, three or more mutations are detected simultaneously in the same plant. There are two types of chimera; (i) the ones with two or more mutated alleles that also present a wild type allele (may introduce knock-down but never complete knock-out) and (ii) the chimera with all alleles mutated that may trigger complete knock-out.

Here, we tested the potential of the CRISPR/Cas9 system to induce gene knock-out in *E. grandis* hairy roots. We first targeted the *EgrCCR1* (*CCR1)* gene because the phenotypes induced by the down regulation of this gene in *E. grandis* hairy roots have been well characterized [14]. We also selected the candidate *Aux/IAA* gene, Egr*IAA9A (IAA9A)*, known to be preferentially expressed in vascular cambium and developing secondary xylem [9]. For CRISPR/Cas9 editing, we used two single guide RNAs (*sgRNAs*) under the control of *Arabidopsis U6* promoter and *Cas9* under the control of the Cauliflower Mosaic Virus *CaMV 35S* promoter in a single vector [30]. We report here that the CRISPR/Cas9 was efficient in generating mutations in both *CCR1* and *IAA9A* in *E. grandis* hairy roots but with very different editing efficiency rates. The phenotyping of *CCR1*-edited lines by FTIR spectroscopy and histochemical analyses confirmed the decreased lignin phenotypes expected in response to *CCR*-downregulation.

## 2. Results

To implement the CRISPR/Cas9 technology in the *Eucalyptus* hairy roots transformation system, we firstly generated the constructs comprising Cas9 protein expression cassette and two single guide RNAs targets for the same gene in a single vector as described in the Materials and Methods section. We then transferred the constructs into *E. grandis* hairy roots using *Agrobacterium rhizogenes.* The transgenic roots were selected by the selection marker of DsRed fluorescence. We then extracted the DNA of transgenic roots and sequenced the corresponding region to reveal the gene editions in the target genes in order to determine whether or not generated the expected knock-out mutants.

### 2.1. Genotyping Revealed High Knock-Down Rate in CCR1 but High Knock-out Rate in IAA9A

To reveal the CRISPR/Cas9 system genome editing and characterize the mutation types, we extracted genomic DNA from *E. grandis* transgenic roots. We then amplified by PCR the target gene regions including the two sgRNA sequences. The PCR amplicons were directly sequenced and analyzed by the web-based tool “Degenerate Sequence Decoding (DSDecode, http://dsdecode.scgene.com). For *CCR1*-lines the majority of the direct sequencing of PCR amplicons could not be read by the DSDecode program. We then subcloned PCR amplicons into pGEM-T vectors and several subclones were sequenced by Sanger sequencing.

#### 2.1.1. High Knock-Down Rate for *CCR1* Editing

The rates of edited plants (hairy roots) were calculated on the basis of the PCR subcloning sequencing results. First, we verified that mutations in either *CCR1* or *IAA9A* genes were totally absent in the control plants transformed with an empty vector harboring the Cas9 cassette without guide RNA sequences (Appendix A). Then, we separated the CRISPR-*CCR1* and *IAA9A* hairy roots into two groups: (i) putative knock-out if all alleles were altered and no WT allele was present and (ii) putative knock-down if a WT allele sequence was still present (Table 1). For instance, a chimera plant present three and more alleles simultaneously with all alleles edited (Altered allele 1/Altered allele 2/Altered allele3, A1/A2/A3, etc.) was considered as a putative knock-out whereas a chimera plant with two or more mutated alleles and one WT allele simultaneous in the same plant (WT/A1/A2, etc.) was considered as a putative knock-down.

For *CCR1*, 100% of the 24 independent transgenic hairy roots (obtained from three independent transformation batches) were edited. All exhibited mutations in at least one allele (Table 1). Ten out of these 24 (41.2%) exhibited monoallelic mutation with a mutated allele and a WT allele (A1/WT). Thirteen out of 24 (54.2%) were chimeric with a WT allele and two or more mutated alleles (WT/A1/A2, etc.; Table 1). Neither homozygote (A1/A1) nor biallelic mutation (A1/A2) was detected. Only one plant was determined as a chimera having editing in all tested alleles (no WT allele) comprising of seven different edited alleles (Table 1, *CCR1*_*22* in Appendix A). One out of the seven editions was a 1-bp substitution generating no change in the CCR1 amino acid sequence and we also could not totally rule out the possibility that this 1-bp substitution was introduced during the PCR step. Whatever the case, this should not lead to a complete *CCR1* knock-out but most likely to a knock-down. 

For *CCR1*, we obtained an allele editing rate of 32.0%, i.e., only 89 among the 278 PCR subclones sequenced exhibited mutated alleles (Table 2). Forty six subclones (51.7%) had mutations in the first sgRNA position, 65 subclones (73.0%) in the second sgRNA position and 22 subclones (24.7%) contained mutations at both sgRNA1 and sgRNA2 positions (Table 2). Noteworthily 19 subclones (21.3%) showed large deletions between sgRNA1 and sgRNA2 as expected when using two guide RNAs separated from approximately 100 bp [30].

The detailed mutations for each edited allele can be found in Appendix A. The two most prevalent edition types were 82 bp deletion (14 alleles, 16%) and 1 bp deletion (11 alleles, 12%), both generating a reading frame shift and premature stop codon as illustrated in Figure 1. Among the 89 edited alleles, 87 generated amino acid changes, potentially modifying the activity of the corresponding enzyme (Appendix A). Noteworthily, the majority (69.7%) of the *CCR1*-targeted edition introduced significant modifications among which 67.4% were reading frame shifts (Table 3) and 24.7% were 15 bp or more insertion/deletions. The very highly complex pattern of *CCR1* edition explained why the DSDecode online tool failed to run, thereby confirming that this analytic tool is not suitable for chimeric or multiple mutations. 

We performed three independent transformation batches for *CCR1*. For the first batch, we collected our samples on 153-day-old roots (109 days of in vitro culture + 44 days in hydroponic culture) in order to obtain roots containing secondary xylem. Although, all the roots were edited, the allele edition rate was low (23.5%; only 50 alleles altered among 213 alleles sequenced; Appendix A). A likely explanation is that some *CCR1* editing events may have impacted too severely roots development as observed previously when down-regulating *CCR1* in hairy roots [14]. Supporting this hypothesis was the dramatic rate of mortality (59.2%) for this first batch of *CCR1* transgenics right after the transfer from in vitro culture to hydroponic medium (Appendix A). This prompted us to verify if the edition rate could be higher if we harvested younger roots cultivated only in vitro without adding the stress of transferring them to hydroponic culture. For the second batch and third batch of transformation CRISPR *CCR1,* we collected the samples at 73 days and 54 days, respectively. The genotyping results clearly showed that the younger the transgenic roots, the higher the edition rate. The editing rate reached 80.8% for the roots collected at 54 days (around 1–2 cm long) and dropped to 46.2% for those sampled at 73 days (Appendix A).

#### 2.1.2. High Knock-Out Rate in *IAA9A* Lines

For *IAA9A*, we obtained 13 independent transgenic hairy roots plants. Twelve out of 13 had editions/mutations in at least one allele, leading to a very high edited plants rate (92.3%; Table 1). In contrast to *CCR1*, most of mutations profiles suggested *IAA9A* knock-out (91.7%; 11 plants out of 12 edited plants). Among them seven plants (58.3%) were biallelic mutations, four plants (33.3%) were chimera with only mutated alleles and no WT allele. One plant (8.3%) had a monoallelic mutation (A1/W). Unexpectedly, one transgenic hairy root had only WT alleles although it exhibited DsRed florescence indicating the presence of the T-DNA. Neither homozygote mutations nor chimera with the WT allele were detected (Table 1).

In total, we sequenced 95 subclones, among which 88 presented mutations, leading to an allele edition rate as high as 92.6%, which was in sharp contrast to the allele edition rate of *CCR1* (32.0%). In the first guide RNA and the second sgRNA positions, 84 (95.5%) and 79 clones (89.8%) had mutations, respectively; whereas 75 clones (85.2%) contained mutations in both sgRNA1 and sgRNA2 positions (Table 2). Thus, eleven out of 13 plants had all alleles altered leading to a putative knock out rate of 84.6%.

The detailed mutations for each altered allele detected can be found in Appendix A. The two most prevalent edition types were 73 bp deletion (26 alleles, 30%) and 1 bp deletion (8 alleles, 9%), both generated reading frame shifts and premature stop codons (Figure 1). Among the 88 edited alleles, 87 generated amino acid sequence changes, thus potentially modifying the properties of the encoded protein (Appendix A). The majority (86.4%) of *IAA9A* targeted edition introduced significant protein modifications (Table 3) including 85.2% of reading frame shifts with premature stop codons, and 56.8% of 15 bp or more indels (insertion/deletions).

In parallel to subcloning, we used the DSDecode online tool on the 13 PCR amplicon sequences (13 transgenic hairy roots). Nine were successfully treated by DSDecode (Appendix A). For three of them, we got the same results as our subcloning and sequencing data (highlighted in green, Appendix A). They included one monoallelic mutation line (*IAA9A_5*), one biallelic line (*IAA9A_15*) and a line without any mutation (*IAA9A_10*). For two of them the results obtained with the two methods were different (highlighted in yellow, Appendix A). For the line *IAA9A_3*, for instance, the analysis of the subcloning results showed that it was a biallelic mutation whereas the DSDecode concluded that it was a homozygote mutation with the allele of 1 bp deletion; for the line *IAA9A_11,* the two edition types detected by subcloning were different from the one edition type and WT allele detected by DSDecode. For the four others, we observed only partial overlapping of the results between the two methods (Appendix A). For example, for the line *IAA9A_1*, the subcloning results showed biallelic mutations (eight out of nine subclones showed a 73 bp deletion and one showed a 6 bp deletion) whereas the DSDecode online tool concluded that it was a homozygote mutation consisting of the allele with a 73 bp deletion. For the four lines failed to be analyzed by DSDecode, the subcloning sequencing revealed one chimeric line (no WT, A1/A2/A3, etc.) and three biallelic mutations (A1/A2).

#### 2.1.3. Mutation Spectra Vary Among sgRNA Targets

Various types of editing were detected in both *CCR1* (43 types) and *IAA9A* (20 types) lines (Appendix A and Supplementary S4), including deletions, insertions and substitutions (Table 4). The most prevalent edition type was deletion (Table 4, Figure 1). For *CCR1*, 52 and 21 alleles exhibited small and large deletions, respectively. In total, 73 out of 89 alleles (82.0%) had deletions. For *IAA9A*, 49 and 38 alleles had small (smaller than 15 bp) and large (larger than 15 bp) deletions, respectively, leading to an overwhelmed majority (87 out of 88 alleles, 98.9%) of the deletion editing type. As expected, using two guide RNAs separated around 100 bp, large deletions occurring between the two sgRNAs were frequently observed, representing 21.3% and 30.7% of the total edition types for *CCR1* and *IAA9A*, respectively. The second most frequent edition type was substitution. For *CCR1* small substitutions represented the second most prevalent type, scoring as high as 32.6% while no large substitutions happened in *CCR1* or *IAA9A* alleles. Insertions were not often seen in *CCR1* edited alleles; only 4 small insertions were detected among 89 altered alleles and no large insertions were observed. In contrast, in *IAA9A* lines, 16 and 12 alleles had small and large insertions, respectively, representing 31.8% (28 alleles out of 88 alleles) of the editing events.

The comparison of the editing rates between the two different sgRNAs for *CCR1*, revealed that 46 and 65 mutations occurred in sgRNA1 and sgRNA2, respectively (Table 2). Using a larger data set of RNAseq from *E. grandis* than the one used for designing the sgRNAs, we detected the presence of a SNP at position 20 flanked by PAM GGG of sgRNA1 (Figure 1, indicated in a yellow character; Appendix A, footnote) that likely explains the different editing rates between the two sgRNAs. Since we used *E. grandis* seeds and not a clone for hairy root transformation, we assumed that those containing a SNP in *CCR1* were not edited at all as shown for 4-coumarate: CoA ligase by [22]. The editing types were also different between the two *CCR1* sgRNAs. Small substitutions as well as insertions were more frequently observed in the sgRNA1 position than in the sgRNA2 position (Table 4). The two sgRNAs of *IAA9A* displayed equivalent editing rates: 84 and 79 alleles had mutations in sgRNA1 and sgRNA2 positions, respectively.

### 2.2. Phenotyping Revealed Expected Alterations of Lignification in CCR1-Edited Lines

The phenotyping was focused on *CCR1*-edited line because we knew the phenotypes of knock-down mutants (previously generated by antisense techniques in our team [14]) to validate our CRISPR/Cas9 technology was successfully implemented in *Eucalyptus* hairy roots. However, we are uncertain of the phenotypes of *IAA9*A-edited lines, and the phenotypic analyses of *IAA9A*-edited lines are ongoing, so it will not be presented here.

#### 2.2.1. Combination of FTIR Spectroscopy and Multivariate Analyses of *CCR1*-Edited Hairy Root Lines

In order to discriminate rapidly and efficiently between the chemotypes of the CRISPR/Cas9 edited-*CCR1* roots (batch 1) and the control ones, we used the Fourier transformed infra-red (FT-IR) spectroscopy, a fast, cheap and non-destructive technique that provides information about the structure of secondary xylem constituents and chemical changes in wood samples [31]. We analyzed all the FTIR spectra obtained by a multivariate statistical tool (here partial least square analysis (PLS-DA)) because such combinations were shown to be powerful to characterize differences between complex biological samples and provide clues concerning the chemical nature of their divergence [32]. As shown in Figure 2A, the two first components of the PLS-DA explained more than 50% of total variability of the samples. The first component (PC1 axis), which explained 42% of the variability clearly separated *CCR1*-edited lines from controls. The second component (PC2 axis) explained different patterns among the *CCR1* transgenic lines.

We used the median of the FT-IR absorption spectra (Figure 2B) and the loadings contribution values to PC1 and/or PC2 to further identify the most discriminant wavenumbers explaining the separation between these contrasting samples. In total, we found 29 discriminant wavenumbers explaining the separation between *CCR1*-edited lines from controls that belong to three main regions of the spectra: 1000–1200 cm^−1^, 1300–1800 cm^−1^ and 2200–3600 cm^−1^ (see Appendix A and Appendix A). Among them 24 reported on FT-IR spectra (Figure 2B) were already described in the literature as bounds related to SCW composition. Notably, the majority of these bands were related to lignin structure and composition (Appendix A). For most of them, the absorption of *CCR1*-edited lines was lower than in controls (Figure 2B) as expected from the down-regulation of a step-limiting enzyme of the lignin biosynthesis pathway.

#### 2.2.2. Histochemical Characterization of *CCR1*-Edited Hairy Root Lines

We further examined the vascular tissues of the transgenic roots by performing histological analyses using either the phloroglucinol-HCL, which stains lignin polymers in red-purple or the natural auto florescence of phenolic compounds (including lignin) under UV-light. In root sections performed at around 10 cm from the root apex, most CRISPR/Cas9 edited *CCR1*-lines displayed a clear *CCR1* down-regulation phenotype: xylem cell walls (vessels and fibers) stained faintly with phloroglucinol-HCl in comparison to roots transformed with control vectors, which appeared strongly stained in red (Figure 3, Appendix A). As the intensity of the phloroglucinol-HCl staining is indicative of the lignin content, the faint staining in the CRISPR/Cas9 edited lines strongly suggest a reduced lignin content. A frequent collapsed xylem vessels and irregular shapes for both xylem vessels and fibers were also detected in those edited lines (Figure 3, indicated by arrows), due to lower lignification, as well as cells with greatly decreased lignification (non-phloroglucinol staining and/or no auto fluorescence under UV light; Figure 3, indicated by *). These observations were consistently obtained from the several lines (Figure 3, Appendix A), especially the strongest phenotypes were more distinct in those such as the lines comprised big deletions (e.g., *CCR1_5*, Figure 3. Appendix A) and the lines had significant mutated alleles (e.g., *CCR1_7*, all edited alleles had shifted reading frame, Appendix A). Under UV-light the intensity of auto-fluorescence was also lower in the CRISPR/Cas9 edited *CCR1*-lines as compared to controls, further supporting the hypolignified phenotypes of the *CCR1*-edited lines.

## 3. Discussion

Whereas some studies have implemented gene editing by CRISPR/Cas9 in woody species including fruit trees [44,45,46,47] and forestry species like poplar [20,22,24,25,29,48] none was yet reported in eucalypts probably because they are particularly recalcitrant to genetic transformation [11,14]. The purpose of this study was to combine the CRISPR/Cas9 gene editing system with the efficient *Eucalyptus grandis* hairy roots transformation system in order to obtain a powerful knock-out system for gene functional studies. We investigated the mutagenesis efficiencies and patterns produced by the CRISPR/Cas9 nuclease directed against two distinct target genes: (i) the lignin biosynthetic gene *CCR1* used previously as a proof-of-concept to show that *Eucalyptus* hairy roots were adapted to functionally characterize SCW-related genes [14] and (ii) the *Aux/IAA* gene *IAA9A*, a potential regulator of xylem formation [9].

For both genes, we obtained very high percentages of edited hairy roots amongst the cotransformed ones, i.e., 100% and 92% for *CCR1* and *IAA9A*, respectively (Table 1). However, the allele-editing rates varied considerably between these two targets. While the allele-editing rate was very high (92.6%) in *IAA9A* transgenic roots, it was low (32.0%) in *CCR1* transgenic roots. Indeed, in *CCR1*-edited transgenic roots, we were surprised by the absence of biallelic editing and by the very high percentage of chimera. In strong contrast, for *IAA9A*-edited transgenic roots, the level of biallelic mutations was high (58.3%) and much less chimera was detected. Moreover, the percentage of potential knock-out among the *IAA9A*-edited lines (11 out of 12 lines) was extremely high, indicating that the CRISPR/Cas9 nuclease editing system used in this study could be highly efficient in *E. grandis* hairy roots. In addition, based on the differences observed between the *IAA9A-* and the *CCR1*-edited hairy roots, the editing efficiency appeared gene- and sgRNA-dependent. The results obtained with *IAA9A* are closer to those obtained in rice [18,49] and poplar [22,29,49,50] where high proportions of biallelic mutations were reported in the T0 transgenic plants but in comparison, we got more chimeric mutations in *IAA9A*-edited hairy roots. Obtaining biallelic mutations in T0 transformants is important for trees, since in poplar for instance, it was shown that biallelic mutations were stably inherited through clonal propagation [22,29,51]. This in agreement with the fact that CRISPR-induced biallelic DNA modifications lead to permanent mutations in edited cells that are inherited mitotically and no more editions are possible.

The phenotypes of the *CCR1*-edited lines analyzed by the combination of FTIR/PLS-DA on one hand, and by histochemistry on the other hand, shared features characteristics of *CCR1*-deficiency reported in other plants such as a low lignin content and collapsed vessels [14,41,42]. We analyzed more in depth the *CCR1*-editing case to understand why we obtained so many chimera transgenic hairy roots likely leading to *CCR1*-knock-down and no biallelic mutations that would lead to *CCR1* knock-out. Indeed, many examples in the literature revealed that too strong *CCR1*-down regulation leads to deleterious effects. In transgenic tobacco plants, the antisense line with the most severely depressed CCR1 activity exhibited dramatic development alterations with reduced size, abnormal morphology of the leaves, collapsed vessels [52]. In poplar down-regulated for *CCR* by sense or antisense strategies, 5% of the transformed plants were dwarf and unable to be acclimatized [53]. Our hypothesis thus is that biallelic *CCR1* editing would be either lethal or would lead to too severely impaired *CCR1*-lines with very poor development that would die prematurely as it was the case for the more severely down regulated *CCR1* transgenic hairy roots obtained by antisense strategy [14]. Since the hairy roots derive from the transformation of one single cell, the chimera may result from monoallelic edition. In this case, as only one allele is edited, the second allele (wild-type) still contains intact sgRNA target sites. During cell division, the Cas9 protein is able to edit the wild-type targets generating a second type of edition and so on and so forth. In the case of CRISPR-edited *CCR1*, we found two types of chimera (i) those with only edited alleles (A1/A2/A3, etc.) and (ii) those still having wild type alleles (WT/A1/A2/A3, etc.). In the former case, the chimera could be considered as stable because all the alleles are altered and no more target sequences are available for Cas9. In the latter case, the transformants are potentially “not stable” since the target sequences (contained in the wild-type allele(s)) are still available for Cas9 editing, especially because *Cas9* is under the control of the constitutive *CaMV35S* promoter. Taking advantage of the three different transformation batches made for *CCR1*, we compared the allele-editing rates at different times after transformation. Indeed, the allele-editing rate was quite high for the younger hairy roots recently transformed with *A. rhizogenes* and decreased rapidly along with the age of the *CCR1*-transformants (Appendix A). One possible explanation is that in chimera hairy roots comprising three or more transformed cell lineages, the severely CCR1-impaired cell lineages were constantly facing the concurrence from surrounding wild-type cell lineages with normal CCR1 function and/or *CCR1* edited cell lineages in which CCR1 function was less impaired. We believed that the severely CCR1-deficient cell lineages would be less competitive and could gradually disappear, leading to a lower allele edition rate in the older transformants. This indirect argument also supports the lethality of a too severe *CCR1*-down regulation and a fortiori of a *CCR1*-knock out and the fact that we did not find any biallelic mutations.

Since the *E. grandis* genome is available [4], we chose the on-line tool ‘CRISPOR’ (http://crispor.tefor.net/) to design the sgRNAs, because it can directly evaluate and score the genome scale off-target risk as well as the editing efficiency [54]. We also selected very low off-target risk sgRNA targets (Appendix A). Although we cannot completely rule out the possibility of having off-targets in our system, most reports in various plants reported a lack of or low percentage of off-target mutagenesis (reviewed by [29]).

In this study, we implemented the CRISPR/Cas9 gene editing system in *Eucalyptus* hairy roots. Among the mutations generated, the majority introduced frame shifts with premature stop codons and thus truncated proteins for both *CCR1* and *IAA9A* genes. In addition, as expected using two gRNAs, large deletions were frequently seen most likely leading to non-functional proteins. The low level of biallelic mutations and the high level of chimera were unexpected especially when compared to poplar [20,22,29], or even to hairy roots system in soybean, tomato or chicory [55,56,57,58,59], where high percentage of biallelic mutations and low number of chimera were reported in general. Although in some studies, the percentage of chimera could have been underestimated by the use of the DSDecode software, which is not able to detect complex patterns of mutations such as chimeric or multiple mutations and is less accurate than subcloning to identify chimera as we showed here. There is room in our system to increase the percentage of biallelic knockout and reduce the mono allelic mutation and chimera percentages. Indeed, for many species, it has been observed that monoallelic/chimera mutations predominate in the first-generation (T0) when CRISPR editing efficiencies were low [19,23]. Multiple parameters were reported impact the edition efficiency such as the use of the native U6 or U3 promoters to drive the expression of the sgRNAs [56,60], the Cas9 expression cassettes. Bruegmann T. and his colleagues [51] also reported that the structure of sgRNA impacts gene editing efficiency, in particular the GC content, the presence of purine residues at the sgRNA end and the free accessibility of the seed region seemed to be highly important for genome editing in poplar. Further studies are needed to explore the impact of the use of (i) *Eucalyptus* native U6 or U3 promoters, (ii) different Cas9 expression cassettes and (iii) optimized sgRNA structures, to achieve higher biallelic edition rate in the T0 generation. This is particularly important for eucalypts that are like poplars, clonally propagated. Although some progress are also needed to improve eucalypts transformation, further research may also be guided to generate CRISPR/Cas9 editing without any transgenic DNA integration by transferring just ribonucleic-protein complexes into plant cells [61] to overcome the persistent societal hostility to transgenic trees.

## 4. Materials and Methods

### 4.1. Plant Material

Commercial *E. grandis* seeds (W. Hill ex Maiden, cultivar LCFA001) purchased at Instituto de Pesquisas e Estudos Florestais (IPEF, Piracicaba, Brazil) were surface-sterilized by 30-min treatment in a 1% sodium hypochlorite solution containing Twin-20. Germination was carried out on 1/4 strength Murashige and Skoog medium (MS medium; Sigma-Aldrich, St. Louis, MO, USA) solidified with 8 g/L (Sigma-Aldrich) at 25 °C in the dark for 3 days. To obtain in vitro plantlets around 1 cm long with the hypocotyls fully expanded and the first two leaves just appearing, we germinated seeds inside plates at normal position at 25 °C for 12 days in light (12 µmol/m^2^/s, 8–16-h photoperiod, 50% humidity).

### 4.2. CRISPR/Cas9 Targeted Mutagenesis System Selection and Pipeline

To implement the CRISPR/Cas9 system in *Eucalyptus*, we selected the method introducing selective marker, 35S-Cas9-Nos expression cassette and two sgRNAs under the corresponding promoter (here *Arabidopsis* U6 promoter) in a single construct that had previously been proven highly active in various plants such as tomato and *Nicotiana benthamiana* [30,62,63] using golden gate cloning [64,65]. The scheme illustrating the pipeline of the targeted mutagenesis in plants is shown in Figure 4, including the main steps as (1) sgRNAs design, (2) construct assembly, (3) hairy roots transformation, (4) genotyping and (5) phenotyping.

### 4.3. CRISPR/Cas9 Target Site Selection and sgRNAs Design

We selected two different target sites for each gene with the help of the sgRNA design online tool ‘CRISPOR’ (http://crispor.tefor.net/), which enables to evaluate the genome wide off-targets and to score on-target efficiency [54]. We first retrieved the genomic sequences of our two target genes obtained from *Eucalyptus grandis* genome sequencing database of Phytozome (https://phytozome.jgi.doe.gov). All possible sgRNAs on the genomic sequence were ranked by CRISPOR based on the off-target risk and on target efficiency. Among those, we selected pairs of sgRNAs at around 100 bp interval to increase editing efficiency and to possibly create a large deletion between the two sgRNAs [62]. The target sites selected had a ‘G’ as their first base to function as the RNA polymerase III start site (Guide RNA prefix for U6 promoter) and were followed by the Protospacer Adjacent Motif (PAM) sequence ‘NGG’ given the *Streptococcus pyogenes* Cas9 as PAM selection preference. The selected sgRNAs and the off-target risk and editing efficiency prediction were presented in Appendix A [66].

Since eucalypts are highly heterozygous, we verified the absence of SNPs in the guide RNAs by blasting them against a large RNAseq data set of *E. grandis* [67] registered at the NCBI SRA database (PRJNA514408). Unexpectedly, the alignment between the RNAseq and the genomic sequence of *CCR1* retrieved from Phytozome (*Eucgr. J03114*) allowed us to detect a deletion of 14 bp (5′-gcttctctcctcgagc-3′) at position 33649876 (Chr. J; Figure 5) in the latter and consequently the predicted exon/intron structure and protein sequence were wrong. We corrected manually the sequence and both gene structure and proteic sequence were in perfect agreement with our previous work [2,68].

Finally, for *CCR1* (Eucgr.J03114), we chose two sgRNAs (sgRNA1_*CCR1* (5′-GCGGTCCAAGCACGAGCACA-3′) and sgRNA2_*CCR1* (5′-GACCGAGTTGGCGTAGGTCT-3′)) separated by 64 bp and both located on the antisense strand in exon 4 (Figure 5). This exon contains a highly conserved region among different *Eucalyptus* species [2]. For *IAA9A* (Eucgr.H02407) we selected a pair of sgRNAs (sgRNA1_*IAA9A* (5′-GTCTCCACCACTTCTGGGTG-3′) and sgRNA2_*IAA9A* (5′-GGCGCCTCTCATGACTGCTT-3′)) located at 54 bp interval in exon 1 (Figure 5). The sgRNA1 and sgRNA2 of *IAA9A* locate 23 bp and 96 bp after ATG start codon respectively.

### 4.4. CRISPR/Cas9 Constructs Assembly

The construct assembly was achieved by two steps using golden gate cloning as described in [62]. In brief, for the first step we generated two intermediary vectors (AtU6p::sgRNA1_*CCR1* and AtU6p::sgRNA2_*CCR1*) by cut-ligation using BsaI endonuclease and T4 ligase. Each vector harbored *Arabidopsis* U6 promoter, corresponding *CCR1*-sgRNA1 or 2 target sequence, sgRNA scaffold and U6 terminator. The primers and sgRNA scaffold used for generating sgRNA intermediary vectors were described in Appendix A. sgRNA scaffold template (pICH86966) and level 0 construct (pICSL01009::AtU6 SpecR), level 1 destination vector (pICH47751 (CarbR) and pICH47761 (CarbR) were provided by Dr. G HU (UMR990 GBF, Toulouse France), which can be ordered from Addgene. The second step in one reaction we cut-ligated all intermediary vectors into one final binary vector pICSL4723 (LB-DsRed-Cas9-sgRNA1-sgRNA2-RB) using BbsI endonuclease and T4 ligase (Figure 6), which includes all the CRISPR/Cas9 components: the plant selective marker DsRed expression cassette (AtUbi10p::DsRed::T35S-terminator) at position 1 flanked to the left border of binary vector, domesticated human codon optimized Cas9 expression cassette (2x35Sp-5′UTR::Cas9::NOS Terminator) at position 2, and two sgRNA expression cassettes (AtU6p::sgRNA::U6terminator) at position 3 and position 4 (Figure 6). The DsRed selection marker vector (AtUbi10p::DsRed::T35S-terminator) was provided by Dr. PM Delaux, UMR5546 LRSV, France. The Cas9 expression cassette vector (pICH47742::35S::Cas9-NOST) and the linker vector (pICH41780 Linker) can be obtained from Addgene plasmids (https://www.addgene.org/) The destination vector pICSL4723 was provided by M. Youles (Sainsbury Laboratory, Norwich, UK). For the empty vector control construct, we cut-ligated the DsRed and 35S-Cas9-NOST expression cassettes but without any sgRNA as described following (Figure 6).

For the first step, the cut-ligation (Type II restriction endonucleases–T4 ligation) reaction (20 μL) was prepared to contain 2 μL BsaI 10× reaction buffer (NEB), 20 U of BsaI (NEB), 20 U of T4 DNA ligase (using high concentration ligase, 20 U/μL, Promega (Charbonnières-les-Bains, France)), approximately 40 fmol insert (100 ng of DNA for a 4 kb plasmid) of pICSL01009::AtU6p vector, sgRNA PCR amplicons harboring targeted sequences (15 ng) and 20 fmol destination vector pICH47751 or pICH47761 (with molar ratio 2:2:1). The reactions were incubated in a thermocycler (marque) for 13 cycles (37 °C, 10 min; 16 °C, 10 min), followed by 20 min of digestion at 37 °C, and 10 min of denaturation at 55 °C. The ligations were then transformed into DH5α *E. coli* (Thermo Fisher (Illkirch-Graffenstaden, France)). White colonies were selected on agar with X-Gal and carbenicillin. *E. coli* PCR, plasmid Miniprep (Promega (Charbonnières-les-Bains, France)), restriction endonuclease digestion and Sanger sequencing were carried out to screen and obtain the assembled level 1 vector (AtU6 promoter and gRNA sequence).

For the second step, a restriction-ligation reaction (20 μL) was set up using T4 ligase buffer (Promega (Charbonnières-les-Bains, France)) plus 2 ng BSA (Promega (Charbonnières-les-Bains, France)), 15 U of BpiI (Thermo Fisher (Illkirch-Graffenstaden, France)), 20 U of T4 DNA ligase, 40 fmol of each insert elements (Vector for DsRed, Cas9, sgRNA1, sgRNA2 and linker) and 20 fmol of destination vector (pICSL4723). The reactions were incubated in a thermo-cycler for 30 cycles (37 °C, 5 min; 16 °C, 5 min), 20 min digestion at 37 °C and 10 min denaturation at 55 °C. The cut-ligation products were transformed in One Shot™ TOP10 Chemically *E. coli* Competent cells (Thermo Fisher (Illkirch-Graffenstaden, France)), and a single white colony was selected (white/orange selection) on LB plates supplemented with kanamycin (100 mg/mL). The colony was grown in liquid culture for 12–16 h 37 °C and the “DsRed-Cas9-sgRNA1-sgRNA2” vector was extracted using a Miniprep kit, ready to transform *Agrobacterium rhizogenes*.

### 4.5. Agrobacterium Rhizogenes-Mediated Transformation

The binary vectors were transferred into *E. grandis* hairy roots using *A. rhizogenes* strain A4RS as described by [14]. *Eucalyptus* composite plants harboring transgenic hairy roots and wild-type shoots were grown in vitro culture (MS with ½ strength of macro elements) for a period of three to ten weeks (7–12 lmol/m^2^/s, 8–16-h photoperiod, 40% humidity, 22/20 °C). In order to obtain roots containing enough xylem, for the first batch of *CCR1* transformation, 109-day-old in vitro culture composite plants were transferred in hydroponic culture using MS with ½ strength of macro elements solution in a phytotron (130 µmol/m^2^/s, 8–16-h photoperiod, 80% humidity, 25/22 °C). After 4 weeks DsRed fluorescence was verified again. For the first batch of *CCR1* transformation, 153 days old hydroponic cultured fluorescent roots expressing DsRed were collected for DNA extraction, chemical analysis (FT-IR) and histochemical analysis. For the second and third batch of *CCR1* transformation, 54 days old and 73 days old in vitro culture florescent roots were sampled for genotyping. For *IAA9A* transformation 165 days old (77 days in vitro culture + 88 days in hydroponic culture) fluorescent roots were samples for genotyping.

### 4.6. DNA Isolation, PCR Amplification and Mutation Identification

Transformed *Eucalyptus* hairy roots (2 cm from the apex) expressing DsRed fluorescence were harvested for genomic DNA extraction by the CTAB method according to [69]. The quality and concentration of the genomic DNA were measured by a Nanodrop (DS-11 Spectrophotometer).

We used PCR to amplify the genomic region flanking the target sites. For *CCR1*, the forward and reverse primers were 102 bp upstream of sgRNA1 and 376 bp downstream of sgRNA2, respectively (*CCR1*_edit check_NCBI_F and *CCR1*_edit check_NCBI_R, amplicon size 579 bp, Appendix A); For *IAA9A* the forward and reverse primers were 372 bp upstream of sgRNA1 and 131 bp downstream of sgRNA2, respectively (*IAA9A*_edit check_NCBI_F, *IAA9A*_edit check_NCBI_R; amplicon size 595 bp; Appendix A). The PCR amplifications were performed using the High-Fidelity Phusion DNA Polymerase (Thermo Fisher (Illkirch-Graffenstaden, France)).

Two methods were used to identify mutations. The PCR amplicons were either directly sequenced and analyzed by the web-based tool “Degenerate Sequence Decoding (DSDecode, http://skl.scau.edu.cn/sadsdecode/#), and/or were subcloned into pGEM-T vectors after adding ‘A’ tail by GoTaq polymerase (Promega (Charbonnières-les-Bains, France)) and up to 22 colonies were sequenced by Sanger sequencing. For the first method, instead of using the primers for previous amplification, two nested primers were used for PCR amplicon directly sequencing followed the instruction of the DSDecode online tool: nested primer *CCR1*_R (located 220 bp downstream of gRNA2) and nested primer *IAA9A*_F (located 191 bp upstream of gRNA1; Appendix A) to avoid the noise signals of sequencing results.

### 4.7. FTIR Analyses

Hairy roots were harvested, frozen in liquid nitrogen and kept at -80 °C until use. Samples were freeze-dried during 48 h and milled with a Mixer Mill MM 400 (Retsch). Fourier transform infrared spectroscopy (FT-IR) analysis was performed on 100-200 mg lyophilized roots dried powder samples using an attenuated total reflection (ATR) Nicolet 6700 FT-IR spectrometer (Thermo Fisher (Illkirch-Graffenstaden, France)) equipped with a deuterated-triglycine sulfate (DTGS) detector. Some *CCR*-edited roots were too small to generate enough material to be analyzed by FT-IR.

Spectra were recorded in the range 400–4000 cm^−1^ with a 4 cm^-1^ resolution and 32 scans per spectrum. We used hyperspectr v0.99 [70], prospect v0.1.3 [71] and base v3.6.2 packages [72] to perform baseline correction, normalization and offset correction, respectively. All packages were compiled with R version i386 3.5.2. Analyses were performed using the mean spectra resulting from ten individual replicates. Partial least square-Discriminant analysis (PLS-DA) was performed using mixOmics R package [73] to compare samples spectra and identify wavenumbers responsible for samples discrimination.

### 4.8. Histochemical Analysis

DsRed fluorescence indicating co-transformed roots was detected using a stereomicroscope Axiozoom V16 (Zeiss, Marly le Roi, France) equipped with a color CCD camera (ICC5; Zeiss) and with filter sets for DsRed (607/80 nm). Transverse sections (60 µm thick) of roots embedded in 6% low gelling point agarose (Sigma-Aldrich) were obtained using a Vibratome (VT 100S; Leica) and observed using an inverted microscope (DM IRBE; Leica) equipped with a CDD color camera (DFC300 FX; Leica). Lignified secondary cell walls were visualized either in red/purple by phloroglucinol-HCl staining or in blue due to auto fluorescence under UV light.

## Figures and Tables

**Figure 1 ijms-21-03408-f001:**
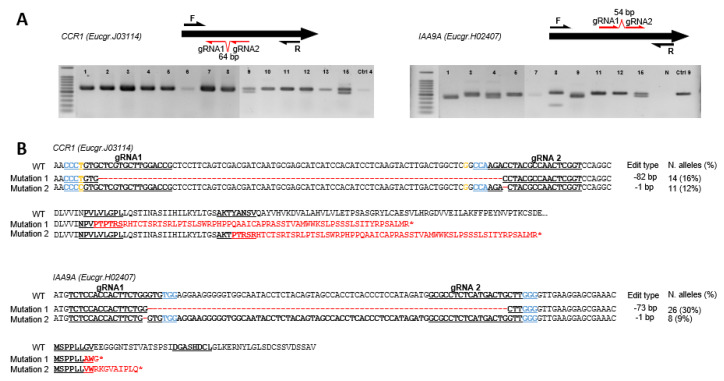
Genotyping of *CCR1* and *IAA9A* transformants and their corresponding prevalent edition types. (**A**) Electrophoresis gels showing PCR amplicons for *CCR1* (left panel) and *IAA9A* (right panel) transformants; the positions of the guide RNAs and of the primers used for PCR are indicated on the schematized ORF (open reading frame) sequences of the genes. The symbols above each lane indicate the transformant lines. In the panel of *CCR1* transformants: 1 for *CCR1_1*, 2 for *CCR1_2* and so on; Ctrl-4, control line 4. In the panel of *IAA9A* transformants: 1 for *IAA9A_1*, 2 for *IAA9A_2* and so on; N, PCR negative control. (**B**) Prevalent mutation types in alleles from transformants *CCR1* and *IAA9A*, respectively. The top alignments show mutations in DNA sequences compared to the WT sequence (red dashes indicate deleted base pairs) and the consequences on the protein sequences are displayed below (altered amino acid sequences are in red). The sequences of sgRNA1 and sgRNA2 are underlined, Protospacer Adjacent Motif (PAM) sequences are indicated in blue. Two SNP positions in *CCR1* are indicated in yellow. The mutation types, number of altered alleles detected and their corresponding occurrence in percentage in brackets are indicated on the right part of the figure.

**Figure 2 ijms-21-03408-f002:**
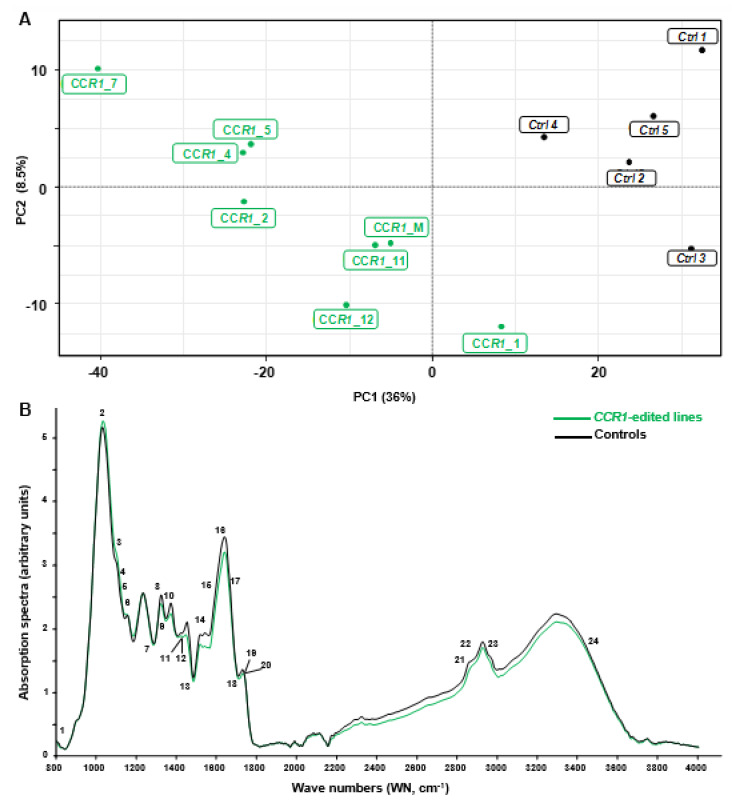
Comparison between FT-IR spectra obtained from controls and *CCR1* edited lines. (**A**) partial least square analysis (PLS-DA) analysis was performed using the normalized values of Fourier-transformed infra-red (FTIR) absorption spectra (800–4000 cm^−1^), acquired from hairy roots samples. The first principal component (PC) separates controls from *CCR1*-edited lines and explains more than 36% of total variability. PC2 axis (8%) mostly explains the separation of different groups within *CCR1*-edited lines. *CCR1*_M represent a mixture of five *CCR1*-edited samples (*CCR1_6, CCR1_8, CCR1_9, CCR1_10* and *CCR1_13*) due to not enough materials if proceeded individually. (**B**) FT-IR absorption spectra of controls (black) and *CCR1*-edited lines (green). The curves were drawn using the median of controls and *CCR1*-edited lines absorption values (except *CCR1-1* and *CCR1-14*). Numbers 1–24 are the most significant wave numbers related to secondary cell wall polymers involved in the separation between controls and *CCR1*-edited lines (see Appendix A and Appendix A) [33,34,35,36,37,38,39,40,41,42,43].

**Figure 3 ijms-21-03408-f003:**
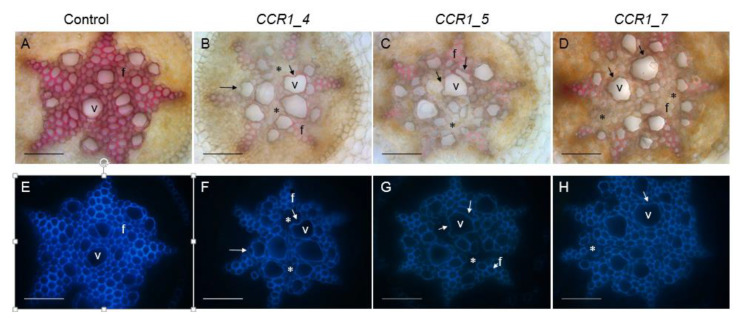
Comparison of xylem development and lignification of xylem cells between control and *CCR1* edited lines’ roots. Transversal root sections made at around 10 cm from the root apex of control (**A**,**E**) and edited lines (**B**–**D**,**F**–**H**). Lignified cell walls are visualized in red/purple by phloroglucinol-HCl (**A**–**D**) and in blue by UV auto fluorescence (**E**–**H**). Collapsed vessels in *CCR1*-edited lines are indicated by arrows. Cells with greatly decreased lignification (non-phloroglucinol staining and/or no auto fluorescence under UV light) are indicated by *. V, vessels; f, fibers. Scale bar = 50 µm.

**Figure 4 ijms-21-03408-f004:**
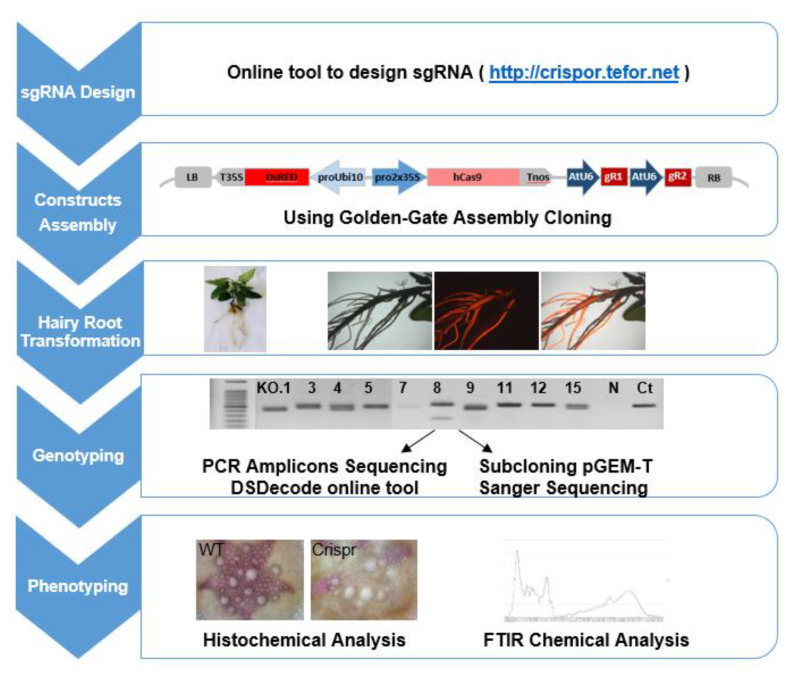
Pipeline of CRISPR/Cas9 implementation in *Eucalyptus grandis* hairy roots.

**Figure 5 ijms-21-03408-f005:**
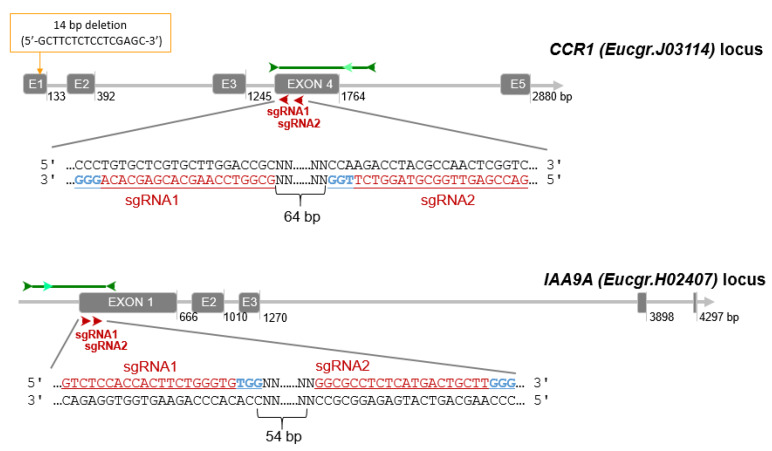
CRISPR/Cas9 sgRNA design and mutation detection in *CCR1* and *IAA9A*. Schematic representation of the target sites and the PCR assay for Sanger sequencing. Exons and introns are represented by gray boxes and gray lines, respectively. The target sites for each CRISPR/Cas9 nuclease are indicated by red arrows, sgRNA target sequence are indicated in underlined red characters, PAM sequences in blue. The dark green arrows indicate approximately the location of the primers for PCR amplification, the light green arrows indicate the nested primers designed for DSDecode mutation identification.

**Figure 6 ijms-21-03408-f006:**
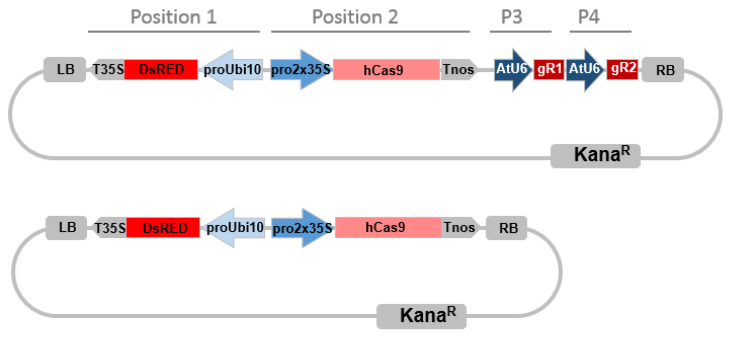
CRISPR/Cas9 binary vector targeting two loci simultaneously. Promoters are indicated in blue arrows, terminators are indicated in gray arrows. *Pro2x35,* double Cauliflower mosaic virus *CaMV 35S* promoter; *AtU6*, *Arabidopsis thaliana U6* gene promoter; *hCas9*, human codon-optimized *Cas9* gene sequence from *Streptococcus pyogenes*; LB, left T-DNA border; *Kana*^R^, kanamycin resistance gene sequence; DsRed, DsRed fluorescent marker gene sequence, *ProUbi10 Arabidopsis thaliana Ubiquitin 10* gene promoter; *T35S*, terminator region of *CaMV35S* gene; *TNos*, terminator region of the *nopaline synthetase* gene from *Agrobacterium tumefaciens*; RB, right T-DNA border, P3, Position 3; P4, Position 4, miss description of the gRNA. The bottom construct is the Cas9 control plasmid without sgRNA.

**Table 1 ijms-21-03408-t001:** Numbers of edited plants and type of mutations according to target genes.

Gene	Edited Plants/Total Transgenic Plants	Plants with All Alleles Altered (no WT)			Plants with one WT Allele		WT/WT
		Homoz. (A1/A1)	Biallelic (A1/A2)	Chimera (A1/A2/A3...)	Monoallelic (WT/A1)	Chimera (WT/A1/A2...)	
*CCR1*	24/24 (100%)	0	0	1 (4.2%)	10 (41.2%)	**13 (54.2%)**	0
*IAA9A*	12/13 (92.3%)	0	**7 (53.8%)**	4 (30.8%)	1 (7.7%)	0	1 (7.7%)

A1, edited allele 1; A2, edited allele 2, WT, wild-type allele, Homoz. Homozygote mutation. Different numbers in the alleles stand for distinct alleles. The number and percentage of the most prevalent genotypes for each CRISPR/Cas9 edited gene is in bold, highlighted in green for *CCR1*-transformants and in yellow for *IAA9A*-transformants.

**Table 2 ijms-21-03408-t002:** Editing frequency and position (identified by PCR subcloning and subsequent sequencing).

Gene	Total Sequenced Subclones	Edited Subclones	Editions in sgRNA1	Editions in sgRNA2	Editions in sgRNA1&2	Large Deletions
*CCR1*	278	89 (32.0%)	46 (51.7%)	65 (73.0%)	22 (24.7%)	19 (21.3%)
*IAA9A*	95	88 (92.6%)	84 (95.5%)	79 (89.9%)	75 (85.2%)	27 (30.7%)

**Table 3 ijms-21-03408-t003:** Majority of altered alleles introduced significant modifications at the protein level.

Gene	Edited Clones	Presumed Significant Modifications		Presumed Less Significant Modifications (15 bp Indel, Substitution without Shift)
		Reading Frame Shift ^1^	≥15 bp Indel	
*CCR1*	89	60 (67.4%)	22 (24.7%)	27 (30.3%)
		62 (69.7%)		
*IAA9A*	88	75 (85.2%)	50 (56.8%)	12 (13.6%)
		76 (86.4%)		

^1^ Reading frame shifts all induced premature stop codons. Indel: insertion and deletion; sub: substitution.

**Table 4 ijms-21-03408-t004:** Mutation types.

Gene	Total Edited Clones	Total Edition Types	Deletion						Insertion						Substitution						Expected Large Deletion
			Small Deletion (≤15 bp)			Large Deletion (>15 bp)			Small Insertion (≤15 bp)			Large Insertion (>15 bp)			Small Substitution (≤15 bp)			Large Substitution (>15 bp)			
			sg1	sg2	sg1&2	sg1	sg2	sg1&2	sg1	sg2	sg1&2	sg1	sg2	sg1&2	sg1	sg2	sg1&2	sg1	sg2	sg1&2	
*CCR1*	89	43	12	40	0	19	21	19	3	1	0	0	0	0	24	5	0	0	0	0	19 (21.3%)
			52 (58.4%)			21 (23.6%)			4 (4.5%)			0			29 (32.6%)			0			
*IAA9A*	88	20	29	39	19	38	27	27	5	11	0	12	0	0	12	2	0	0	0	0	27 (30.7%)
			49 (55.7%)			38 (43.2%)			16 (18.2%)			12 (13.6%)			14 (15.9%)			0			

sg1, sgRNA1; sg2, sgRNA2; sg1&2, sgRNA1 and sgRNA2. The most prevalent edition type is highlighted in green and the second most prevalent types are highlighted in yellow for both *CCR1* and *IAA9A* alleles.

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
