# Peer review of "Implementing the CRISPR/Cas9 Technology in *Eucalyptus* Hairy Roots Using Wood-Related Genes"

_ijms, 2020, doi:10.3390/ijms21103408_

Round 1

Reviewer 1 Report

Review of the paper by Y. Dai and colleagues “Implementing the CRISPR/Cas9 Technology in Eucalyptus Hairy Roots Using Wood-related Genes”

for IJMS; Manuscript ID: ijms-788879.

Reviewer's report

The present paper reports the implementation of the powerful CRISPR/Cas9-based gene editing technology with the hairy roots transformation system in a commercially important tree species Eucalyptus grandis. The CRISPR/Cas9 gene editing in forest trees is very interesting, because this technology can accelerate forest tree breeding in an undreamed manner.

I highly recommend to publishing this manuscript, however, some minor comments should be considered before:

  • Abstract, Lines 15-30: the authors mention that two genes, CCR1 and IAA9A, are selected for CRISPR/Cas9 editing, however, they describe just the results for CCR1. Further, I recommend to make clear that knock-out mutants are aimed following CRISPR/Cas9 editing.
  • Lines 66-67: the abbreviation for CRISPR/Cas9 is not totally correct: Clustered Regularly Interspaced Short Palindromic Repeats / CRISPR-associated protein 9. Of course, Cas9 is a endonuclease
  • Lines 102-115: the described editing cases are correct but are these results? Line 100 clearly says that results will be described. I recommend to move this paragraph in the introduction section
  • Line 116: why do the authors differentiate between “Gene-edition Rate in CCR1” and “Knock-out Rate in IAA9A”? This is not clear at this point of the manuscript.
  • Lines 118-134: again: the description of methods to characterize CRISPR/Cas9 mutations doesn’t contain any result. I recommend to move this paragraph e.g. somewhere in the discussion section
  • Lines 143/144, 154/155: for me, it’s not totally clear what “two or more mutated alleles and one WT allele (WT/A1/A2…)” does mean? Ok, I understand that when 5 plants are obtained, all with one WT-allele, and the first plant with a second allele with mutation 1 (A1/WT), a second plant with a second allele with mutation 2 (A2/WT), the third plant with a second allele with mutation 3 )A3/WT), etc., then this is clear to me. But then I don’t understand the difference to the heterozygote (A1/WT) plants.
    Or do the authors mean that the second allele carries more than one mutation?
    Especially the sentence in lines 154/155 “Thirteen out of 24 (54.2%) were chimeric with a WT allele and two or more mutated alleles (WT/A1/A2…) (Table 1)” implies that each edited plant carry more than two alleles.
    This has to make clear.
  • Line 404: please add link for the ‘CRISPOR’ tool, as it has been done in line 456
  • Many times in the manuscript: please add “City” and “Country” in brackets behind company names, e.g. “Thermo Fisher” or “Promega” (lines 529, 534, 538, etc.). Spelling mistake “Thermo Fisher” in line 534.
  • Line 543 and 590: it’s unlucky that a headline appears at the bottom of the page and the following text on the next page

Author Response

Answer to reviewer1:

The present paper reports the implementation of the powerful CRISPR/Cas9-based gene editing technology with the hairy roots transformation system in a commercially important tree species Eucalyptus grandis. The CRISPR/Cas9 gene editing in forest trees is very interesting, because this technology can accelerate forest tree breeding in an undreamed manner.

I highly recommend to publishing this manuscript, however, some minor comments should be considered before:

Abstract, Lines 15-30: the authors mention that two genes, CCR1 and IAA9A, are selected for CRISPR/Cas9 editing, however, they describe just the results for CCR1. Further, I recommend to make clear that knock-out mutants are aimed following CRISPR/Cas9 editing.

We agree with the comment of reviewer 1 to make clearer in the abstract the objective of generating knock-out mutants using CRISPR/Cas9 editing (see lines 17-18)

We do not completely agree with the fact that we described only the results for CCR1. We genotyped both edited CCR1 and IAA9A lines and this is reported in the abstract (See lines 21-24:  Almost all transgenic hairy roots were edited but the allele-editing rates and spectra varied greatly depending on the gene targeted. Most edition events generated truncated proteins, the prevalent edition types were small deletions but large deletions were also quite frequent). Most of the studies about CRISPR/Cas9 focus only on genotyping of CRISPR-induced mutations. Here we went a step further by analyzing the phenotypes of the CCR1-lines because the phenotype of CCR1 down-regulation is well documented in the literature and well-known in our lab. The phenotyping was also important because it suggested that the absence of knock-out for CCR1 was likely due to the lethality of the KO roots. We did not present IAA9A knock-out phenotypes since the phenotyping is ongoing and will take time since we do not have any a priori idea of the phenotype except that it is likely related to xylem formation.

We made clearer in the revised manuscript that our phenotyping results were focused on the CCR1 lines because we knew the phenotype and that the phenotyping of IAA9A edited roots was ongoing and therefore not be presented in this manuscript (lines 271-275).

Lines 66-67: the abbreviation for CRISPR/Cas9 is not totally correct: Clustered Regularly Interspaced Short Palindromic Repeats / CRISPR-associated protein 9. Of course, Cas9 is a endonuclease
Your comment is relevant and we have corrected this in the revised manuscript (line 67).

Lines 102-115: the described editing cases are correct but are these results? Line 100 clearly says that results will be described. I recommend to move this paragraph in the introduction section.
We agree with this comment and have removed this paragraph in the introduction section (lines 89-99).

Line 116: why do the authors differentiate between “Gene-edition Rate in CCR1” and “Knock-out Rate in IAA9A”? This is not clear at this point of the manuscript.
The reviewer is right, we have changed the sentence in the revised manuscript as follows: “Genotyping Revealed High Knock-down Rate in CCR1 but High Knock-out Rate in IAA9A” (line 119) 

Lines 118-134: again: the description of methods to characterize CRISPR/Cas9 mutations doesn’t contain any result. I recommend to move this paragraph e.g. somewhere in the discussion section.
We agree with your comment and have changed this accordingly.

Lines 143/144, 154/155: for me, it’s not totally clear what “two or more mutated alleles and one WT allele (WT/A1/A2…)” does mean? Ok, I understand that when 5 plants are obtained, all with one WT-allele, and the first plant with a second allele with mutation 1 (A1/WT), a second plant with a second allele with mutation 2 (A2/WT), the third plant with a second allele with mutation 3 )A3/WT), etc., then this is clear to me. But then I don’t understand the difference to the heterozygote (A1/WT) plants.
Or do the authors mean that the second allele carries more than one mutation?
Especially the sentence in lines 154/155 “Thirteen out of 24 (54.2%) were chimeric with a WT allele and two or more mutated alleles (WT/A1/A2…) (Table 1)” implies that each edited plant carry more than two alleles.
This has to make clear
.
Eucalypts are diploid, we were also initially expected to detect either mono-allelic mutation (wild-type allele/altered allele1, WT/A1, also referred as heterozygote), or biallelic mutation (homozygous mutation A1/A1, or biallelic mutation A1/A2). However, unexpectedly we obtained plants presented three or more altered alleles and wild-type allele in the same plant which correspond to chimera. We thus followed the description of chimera (WT/A1/A2) from a published paper from Professor Strauss’ team “Variation in mutation spectra among CRISPR/Cas9 mutagenized poplars” (Elorriaga et. al. 2018), and described them as “two or more mutated alleles and one WT allele (WT/A1/A2..).

WT/A1/A2 describes the chimera plant which presents 3 alleles simultaneously in the same plant: wild-type allele and altered allele A1, another type of altered allele A2 (the alteration of sequence in A2 is different from that in A1). Different numbers in the subscript of the alleles stand for distinct alleles.

Another example: WT/A1/A2/A3 describe the plant present 4 alleles in the same plant, WT, and altered allele 1 (A1), altered allele 2 (A2) and altered allele 3 (A3).

You criticisms is relevant and we totally accepted. In order to make it clearer we modified accordingly the description in the Results section and the Table 1 (Lines 132-150).  

Line 404: please add link for the ‘CRISPOR’ tool, as it has been done in line 456
Many times in the manuscript: please add “City” and “Country” in brackets behind company names, e.g. “Thermo Fisher” or “Promega” (lines 529, 534, 538, etc.). Spelling mistake “Thermo Fisher” in line 534.
This has been changed now according to your comment. We added in the revised manuscript the link for the ‘CRISPOR’ toll, and we added the City and County in brackets behind company “Thermo Fisher”, “Promega”. We also corrected the spelling mistake “Thermo Fisher” in line 534.

Line 543 and 590: it’s unlucky that a headline appears at the bottom of the page and the following text on the next page
Thanks for your careful reading. With the revision, these headlines are not any more at the bottom of the pages and the following text on the next pages.

Reviewer 2 Report

In the current study, authors combined the powerful hairy root transformation method in Eucalyptus with the versatile CRISPR/Cas9 gene editing. This proof-of-concept study showed that these two methods can be combined and used successfully for generating gene knockouts and study the functions of genes in Eucalyptus. It is most certainly will be very useful in the future for plant scientist.

Major comments:

  • Authors used the CRISPR/Cas9 gene editing to knockout both CCR1 gene and also IAA9 gene. While they present phenotypes of CCR1, there is not any results given for the edited plants for IAA9 in terms of phenotype. What phenotypes did the authors see? Or were there no phenotypes? Nevertheless this should be explained in the text.

  • Authors should check the expression of CCR1 and IAA9 genes in their respective edited plant lines, to show that indeed they are knockdown or knock out plants.

  • Finally while authors discuss off-target frequency is likely low, however they did not do any sequencing to show that.

Minor comments:

In the beginning of the results they just jump to characterize the CRIPS/Cas9 mutations, maybe they should write a few sentences to introduce what they have done. It would improve the flow of the article.

Also some of the citations can be corrected.

Author Response

In the current study, authors combined the powerful hairy root transformation method in Eucalyptus with the versatile CRISPR/Cas9 gene editing. This proof-of-concept study showed that these two methods can be combined and used successfully for generating gene knockouts and study the functions of genes in Eucalyptus. It is most certainly will be very useful in the future for plant scientist.

Major comments:

Authors used the CRISPR/Cas9 gene editing to knockout both CCR1 gene and also IAA9 gene. While they present phenotypes of CCR1, there is not any results given for the edited plants for IAA9 in terms of phenotype. What phenotypes did the authors see? Or were there no phenotypes? Nevertheless this should be explained in the text.

Indeed we presented the genotyping results for both genes but we “phenotyped” only the CCR1-edited lines. Most of the studies about CRISPR/Cas9 focus only on genotyping of CRISPR-induced mutations. Here we went a step further by analyzing the phenotypes of the CCR1-lines because the phenotype of CCR1 down-regulation is well documented in the literature and well-known in our lab. In fact, we targeted first CCR1 gene as a proof of concept because we knew the phenotype. Since we obtained only knock-down and no complete knock-out line for CCR1, we tested another candidate gene (IAA9A) possibly involved in wood formation. This enabled us to show that we could obtain complete knock-out using the CRISPR technology combined with the hairy root transformation system. The phenotyping was also important because it suggested that the absence of knock-out for CCR1 was likely due to the lethality of the KO roots. We did not present IAA9A knock-out phenotypes since the phenotyping is ongoing and will take time since we do not have any a priori idea of the phenotype except that it is likely related to xylem formation. We expect wood–related phenotypes confirming our hypothesis of IAA9A being an auxin dependent wood formation regulatory gene but it is also possible that no visible phenotypes occur in our experimental conditions. Whatever the results, they will not add much to this manuscript whose goal was essentially to show that the CRISPR/Cas9 technology could efficiently edit targeted genes (genotyping).

We made clearer in the revised manuscript that our phenotyping results were focused on the CCR1 lines because we knew the phenotype and that the phenotyping of IAA9A edited roots was ongoing and therefore not be presented in this manuscript (lines 271-275).

Authors should check the expression of CCR1 and IAA9 genes in their respective edited plant lines, to show that indeed they are knockdown or knock out plants.

We did not measure transcripts abundance because gene editions generated by CRISPR/Cas9 technology may not change transcripts’ abundance but instead alterations of the transcript sequences might affect gene translation. The majority of the mutations obtained in this study introduced shifted reading frames and premature Stop codons. To the best of our knowledge, transcript abundance is generally not evaluated in the literature dedicated to the CRISPR/Cas9 technology.

Finally while authors discuss off-target frequency is likely low, however they did not do any sequencing to show that.
It’s true that we did not perform any whole genome sequencing because the costs were too high for an objective beyond that of this paper that was only a proof of concept that the Crispr cas9 could work efficiently in eucalyptus hairy roots. We agree with you that this kind of verification will be necessary in future studies to optimize the technique in Eucalyptus hairy roots.

Minor comments:

In the beginning of the results they just jump to characterize the CRIPS/Cas9 mutations, maybe they should write a few sentences to introduce what they have done. It would improve the flow of the article.

We have now provided a paragraph in the beginning the results to introduce what we have done to improve the flow of the article (lines 112-118). Thanks for your pertinent suggestion.

Also some of the citations can be corrected.

We revised the citations through of the whole manuscript now.